

# Use of the melting curve assay as a means for high-throughput quantification of Illumina sequencing libraries

Hiroshi Shinozuka[1,2] and John W. Forster[1,2,3]

[1] Department of Economic Development, Jobs, Transport and Resources, Biosciences Research Division, AgriBio, Centre for AgriBioscience, Bundoora, Victoria, Australia
[2] Dairy Futures Cooperative Research Centre, Australia
[3] School of Applied Systems Biology, La Trobe University, Bundoora, Victoria, Australia

## ABSTRACT

**Background.** Multiplexed sequencing is commonly performed on massively parallel short-read sequencing platforms such as Illumina, and the efficiency of library normalisation can affect the quality of the output dataset. Although several library normalisation approaches have been established, none are ideal for highly multiplexed sequencing due to issues of cost and/or processing time.

**Methods.** An inexpensive and high-throughput library quantification method has been developed, based on an adaptation of the melting curve assay. Sequencing libraries were subjected to the assay using the Bio-Rad Laboratories CFX Connect™ Real-Time PCR Detection System. The library quantity was calculated through summation of reduction of relative fluorescence units between 86 and 95 °C.

**Results.** PCR-enriched sequencing libraries are suitable for this quantification without pre-purification of DNA. Short DNA molecules, which ideally should be eliminated from the library for subsequent processing, were differentiated from the target DNA in a mixture on the basis of differences in melting temperature. Quantification results for long sequences targeted using the melting curve assay were correlated with those from existing methods ($R^2 > 0.77$), and that observed from MiSeq sequencing ($R^2 = 0.82$).

**Discussion.** The results of multiplexed sequencing suggested that the normalisation performance of the described method is equivalent to that of another recently reported high-throughput bead-based method, BeNUS. However, costs for the melting curve assay are considerably lower and processing times shorter than those of other existing methods, suggesting greater suitability for highly multiplexed sequencing applications.

Corresponding author
Hiroshi Shinozuka,
hiroshi.shinozuka@ecodev.vic.gov.au

## INTRODUCTION

Substantial reductions in both the cost and processing time of DNA sequencing have been achieved in the last decade, due to improvements of massively parallel short-read sequencing technologies, and this trend is expected to continue for the next several years or more (http://ark-invest.com/genomic-revolution/declining-costs-of-genome-sequencing). As a consequence, sequencing library preparation procedures, rather than the DNA sequencing

**Table 1  Cost and processing duration assumptions for 92–96 samples using the library normalisation or quantification methods.** SPRI denotes the SPRI bead-based DNA size-selection and purification before the quantification procedure. Size-specificity indicates that the method is able to exclude both unnecessary long (e.g., >1 kb) and short (e.g., <300 bp) DNA (++), or merely short DNA (+) from quantification. Library-specificity indicates that the method detects only dsDNA with the sequencing adaptors on both ends (+), or also dsDNA without the sequencing adaptor(s), from which clonal sequence clusters cannot be generated (−). Expenditure on instruments is not included in the per-sample cost. More details can be found in Table S1.

| Method | Method type | Size specificity | Library specificity | Per-sample cost (US$) | Processing time for 92–96 samples | Reference |
|---|---|---|---|---|---|---|
| SequalPrep | Normalisation | ++[a] | − | 0.91 | 1 h 30 min | *Campbell, Harmon & Narum (2015)*, *Harris et al. (2010)* |
| BeNUS | Normalisation | ++ | − | 1.04 | 1 h 30 min | *Hosomichi et al. (2014)* |
| qMiSeq | Quantification | ++ | + | ~10 | c. 1 day | *Katsuoka et al. (2014)* |
| TapeStation | Quantification | ++ | − | 2.75 | 2 h 15 min | *Kong et al. (2014)* |
| SPRI + real-time PCR | Quantification | + | + | 1.41 | 4 h 50 min | *Buehler et al. (2010)*, *Rohland & Reich (2012)* |
| SPRI + NanoDrop | Quantification | + | − | 0.48 | 1 h 50 min | *Harris et al. (2010)* |
| MC assay | Quantification | + | − | 0.15 | 40 min | |

**Notes.**
[a] DNA fragments between 100 bp and 20 kb are purified with the SequalPrep kit (Thermo Fisher Scientific).

process itself, provide the major bottleneck for large-scale experimentation (*Rohland & Reich, 2012*). As a cost-effective approach, single sequencing libraries are parallel-processed from individual samples, and assigned a unique oligonucleotide index. The libraries are then pooled for multiplexed processing in a single run (or lane) of a high-throughput sequencing platform. During this pooling step, equalisation of DNA quantity from each library is essential for an optimised sequencing output (*Hosomichi et al., 2014*).

Although several library normalisation methods have been established, all of them incur a relatively high cost and/or require long processing times for a large number of samples (Table 1, Table S1) (*Buehler et al., 2010*; *Campbell, Harmon & Narum, 2015*; *Harris et al., 2010*; *Hosomichi et al., 2014*; *Katsuoka et al., 2014*; *Kong et al., 2014*; *Rohland & Reich, 2012*). Single libraries are commonly quantified with a well-established real-time PCR-based method (*Buehler et al., 2010*; *Rohland & Reich, 2012*) which, however, requires a relatively long processing time. The qMiSeq method was recently proposed as a highly accurate library titration method for multiplexed sequencing libraries (*Katsuoka et al., 2014*). In this method, the pooled library is first sequenced on the MiSeq instrument (Illumina, CA, USA) in order to estimate the quantity of single libraries, which are then re-pooled for sequencing on the HiSeq platform (Illumina). Although this method is highly efficient when applied to an extensive sequencing experiment for an organism with a large genome, the per-sample cost approaches US$10 if the sample number is 93 or equivalent.

Melting curve (MC) assays identify the melting temperature (MT) point at which double-stranded DNA (dsDNA) dissociates into single-stranded DNA (ssDNA), through detection of reductions in dsDNA-specific fluorescence signals during the heating process (*Ririe, Rasmussen & Wittwer, 1997*). This assay can be performed on most real-time PCR instruments, and up to 96–1,536 samples can be simultaneously processed, depending on the instrument format (*Lennon et al., 2010*). MT largely depends on the length of

dsDNA, such that shorter dsDNA molecules, in general, dissociate at lower temperatures than longer ones (*Ririe, Rasmussen & Wittwer, 1997*). As a consequence, the MC assay is commonly used for quality control purposes in real-time PCR experiments that show presence/absence of undesirable short amplicons, such as PCR primer dimers.

An application of the MC assay for quantification of sequencing libraries is reported in this study. Libraries for multiplexed sequencing are generally enriched through PCR, and short DNA molecules, such as PCR primers, primer dimers, and amplicons derived from self-ligated sequencing adapters must be excluded from quantification. Although SPRI (solid phase reversible immobilisation)-based DNA size-selection is commonly used for this purpose (*Harris et al., 2010*), this approach requires a relatively large amount of consumables (micropipette tips and microtiter plates) and a long processing time. This procedure is not required for MC assay-based quantification, which permits exclusion of such short DNA molecules from library quantification, based on differences of MT. In addition, as real-time PCR instruments are commonly present in molecular biology laboratories, this normalisation approach can be readily applied in labs and hence may not require the acquisition of a new instrument.

## MATERIALS & METHODS

### Preparation of short DNA and KAPA fragments

In order to generate short DNA amplicons (<300 bp), which are normally eliminated from sequencing libraries, an in-house library procedure was performed in the absence of DNA template. An in-house-designed double-stranded Y-shaped adaptor (Fig. S1) with a single thymine base extension (50 pmol) was treated with the Klenow fragment of *Escherichia coli* DNA polymerase I (conferring $3' \rightarrow 5'$ exonuclease activity) (New England Biolabs; NEB, MA, USA) in the presence of dATP, followed by self-ligation using T4 DNA ligase (NEB). The ligated products were purified with Lambda Exonuclease and Exonuclease I (NEB), and then subjected to PCR amplification for 32 cycles with the Phusion DNA polymerase kit (Thermo Fisher Scientific, MA, USA) and in-house library preparation primers containing sequencing indexes. PCR amplification was also performed using the DNA standard 3, primer mix and qPCR master mix of the KAPA Library Quantification Kit Illumina Platforms (Kapa Biosystems, MA, USA), to generate amplicons of 452 bp in length. Sample mixtures (total volume: 20 µl), containing the short DNA or KAPA fragments, 1× PCR buffer, and SYBR Green I (Thermo Fisher Scientific), were prepared for the MC assay.

### Preparation of sequencing libraries

Sequencing libraries for the MiSeq platform were prepared following a previously described high-throughput method based on use of the methylation-dependent restriction endonuclease MspJI (NEB) (*Shinozuka et al., 2015*). Genomic DNA of strains of the most prevalent perennial ryegrass-associated fungal endophyte (*Epichloë festucae* var. *lolii*), of which genome size is around 30 Mb, was directly amplified from mycelium using the REPLI-g Mini Kit (QIAGEN, Hilden, Germany). In the amplification solution of the kit, 16.7 µM 5-methylcytosine (TriLink biotechnologies, CA, USA) was included. The

amplicons were digested with MspJI in order to promote semi-random DNA fragmentation. The DNA fragments were treated with the Klenow fragment, followed by ligation of the in-house-designed adaptor using T4 DNA ligase. The DNA was then purified and size-selected using AMPure XP beads (Beckman Coulter, CA, USA), a portion of which was used for PCR enrichment with the in-house-designed primers using the Phusion polymerase kit. During this process, a unique sequencing string was attached to each sample. Using SYBR Green I as a fluorescence dye, amplification was monitored on the CFX Connect™ Real-Time PCR Detection System (Bio-Rad Laboratories, CA, USA).

## MC assay-based dsDNA quantification

For the MC assay-based quantification procedure, 10 µl of DNA staining mixture, containing 0.1 µl 100× SYBR Green I, 2 µl 5× PCR buffer, 1.5 µl 500 mM EDTA and 6.4 µl PCR-grade water, was prepared and added to 20 or 25 µl PCR products. The MC assay was performed on the real-time PCR instrument. Initial sample heating was performed at 75 °C for 30 sec, and then samples were incubated at each degree point for 5 or 10 s, followed by regular temperature increments of a single degree. Fluorescence measurement was performed after incubation at each degree point. The library concentration was calculated through summation of reduction of relative fluorescence units (dRFU) between 86 and 95 °C.

## Automated gel electrophoresis-based DNA quantification

Library quantification was performed with the 2200 TapeStation Instrument and the D1000 kits (Agilent Technologies, CA, USA). In order to reduce technical error, 2 µl of PCR product was added to 6 µl D1000 Reagent, instead of 1 µl DNA sample and 3 µl D1000 Reagent as described in the manufacturer's instruction. Quantity of DNA between 300–700 bp in length was manually determined and expressed as library quantity.

## NanoDrop-based DNA quantification

PCR products were purified with 0.8 times volume of AMPure XP beads in order to remove DNA fragments shorter than 300 bp in length, and DNA was eluted in 10 mM Tris–HCl buffer. The concentration of purified DNA was measured on the NanoDrop 1000 instrument (Thermo Fisher Scientific).

## Massively-parallel short-read sequencing

Sequencing libraries were pooled based on the basis of quantification result. The pooled library was purified using 0.8 times volume of AMPure XP beads, and subsequently characterised with the TapeStation and Qubit instruments (Thermo Fisher Scientific). The pooled library was loaded and sequenced on the Illumina MiSeq platform, following the manufacturer's instruction. Output data were analysed using the PRINSEQ-Lite software (*Schmieder & Edwards, 2011*; http://prinseq.sourceforge.net/). The percentage of paired-end read numbers (PRN) among 66–75 indexes was calculated for evaluation of normalisation performance (*Hosomichi et al., 2014*). The normalisation degree at 50% ($ND_{50}$) was calculated as the ratio of single libraries of which the PRNs were 50% or more of the average PRN.

## RESULTS

### Determination of a temperature range for MC assay

The short DNA or KAPA fragments were subjected to the MC assay. The results indicated a large portion of the short fragments, of which the majority were less than 300 bp in length, dissociated into ssDNA at temperatures less than 86 °C, while the 452 bp fragment only began to dissociate at 86 °C (Figs. S2A and S2B). The short DNA and KAPA fragments were also subjected to the assay in two different types of PCR buffers, demonstrating that components of the buffers slightly affected the MT of both DNA fragments (Fig. S2C).

### Comparison of the library quantification methods

Using parallel-processed sequencing libraries, quantification results of the MC assay-based method was compared with those of TapeStation-, and NanoDrop-based methods (Fig. S3). The sequencing libraries were prepared from 67 *Epichloë festucae* var. *lolii* endophyte samples. Following adaptor ligation, the libraries were enriched through PCR. Monitoring of the amplification process on the real-time PCR instrument revealed that most samples reached the PCR plateau phase within 14 reaction cycles. Using another portion of the adaptor-ligated products, PCR amplification was performed for 12 cycles in the absence of fluorescence dye, to generate sequencing libraries with a range of dsDNA concentration. The concentration of the 12-cycled products was measured using each of the three methods: MC assay-, TapeStation-, and NanoDrop-based quantification (Fig. 1). The DNA library concentrations varied from 0 (undetectable) to 7.51 ng/µl, obtained with the TapeStation-based method. The results from the three methods were correlated with one other ($R^2 > 0.77$). Comparison of the three methods indicated that a total dRFU value of 100 may be sufficient for relatively reliable quantification when the MC assay-based method is used. The sum of dRFU values between 75 and 95 °C exhibited a lower correlation with the other quantification results (Fig. S4).

### MC-based library quantification and normalisation

The 14-cycled libraries were subjected to the MC assay (Fig. 2), revealing that the total dRFU values between 86 and 95 °C varied less than those of the 12-cycled samples. A value from a single sample was negative (−5), indicating failure of library preparation. The sum of dRFU from the rest of samples was over 150. The 66 successful libraries were pooled for multiplexed sequencing, according to the quantification result (Table S2). From the pooled library, 4.7 million paired-end reads were generated using a part of a sequencing run on the MiSeq platform. The PRNs of the 66 libraries varied from 0.45 to 3.21, with an average of 1.52, a standard deviation of 0.47 and coefficient of variation (CV) of 0.31 (Fig. 2). Except for the lowest library, the PRNs were over 0.77, and the $ND_{50}$ was consequently 0.98. Based on the obtained sequencing data, PRNs in the absence of a normalisation procedure were also predicted. This simulation revealed that the PRNs from 8 libraries ($8/66 = 0.12$) would be less than half of the average ($1.52/2 = 0.76$), if an equal volume of libraries were pooled. The CV of PRNs for the un-normalised pool was calculated to be 0.38.

For further validation, sequencing libraries were prepared from another set of 75 *Epichloë festucae* var. *lolii* strains. Following library enrichment based on PCR, single libraries were

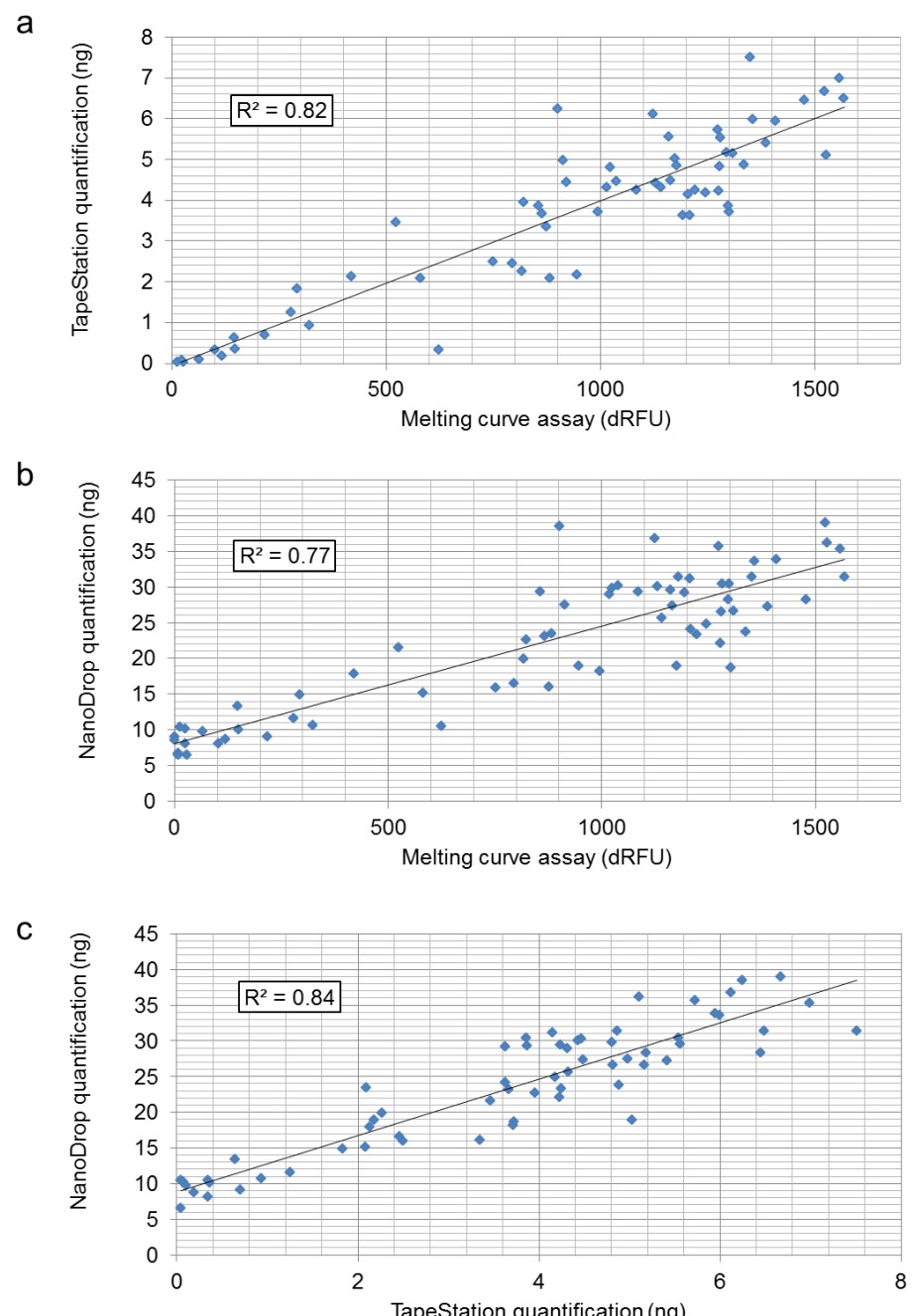

**Figure 1** Correlation analysis between the three library quantification methods: (A) MC assay- and TapeStation-based methods; (B) MC assay- and NanoDrop-based methods; and (C) TapeStation- and NanoDrop-based methods.

subjected to the MC assay-based quantification procedure (Fig. S3). The total dRFU values from libraries varied from 20 to 3,537 (Fig. 3A), while that of the no-template PCR control was 15. From each library, the same volume (5 μl) of the PCR products was pooled and then purified with the AMPure XP beads. The pooled library was loaded on the MiSeq platform, and a total of 1.4 million reads were generated using a proportion of a MiSeq

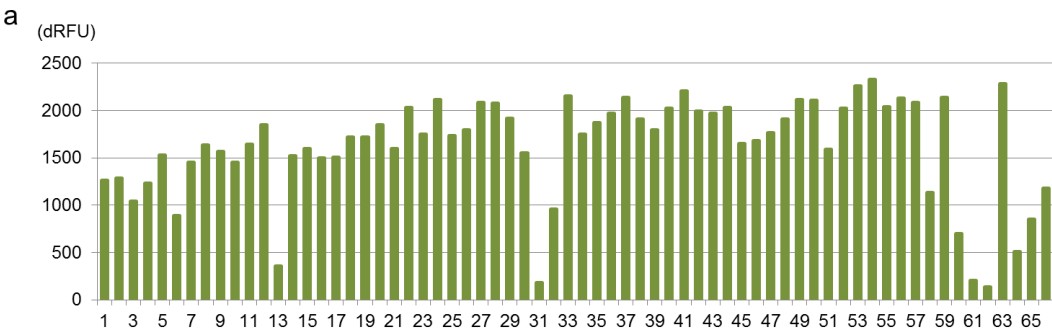

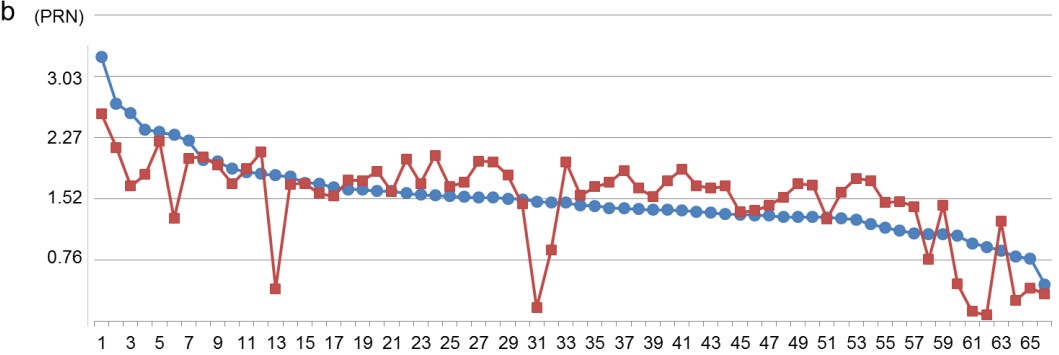

**Figure 2** **Results of the MC assay (A) and high-throughput sequencing (B) from the 66 successful libraries.** The horizontal axis shows the sample unique identifier. In the result from high-throughput sequencing, the blue circle indicates PRN from the MiSeq-derived output, and the red square shows the simulated PRN for un-normalised multiplexed sequencing.

sequencing run. The PRNs of the sequencing output were correlated with the total dRFU values ($R^2 = 0.82$). A further correlation analysis indicated that the total dRFU values for the temperature range between 85 and 95 °C were most strongly correlated to the PRNs ($R^2 = 0.823$), followed by the total dRFU values between 85 and 94 °C ($R^2 = 0.822$), and between 86 and 95 °C ($R^2 = 0.821$) (Fig. S5).

A simulation study was also performed using these results. Assuming that the total dRFU values between 86 and 95 °C are in direct proportion to library concentrations, volumes of the PCR products for pooling were calculated. As the total dRFU values from two of the libraries were lower than 100 (20 and 26), these were excluded from the simulation. The simulated PRNs were obtained based on the calculated pooling volume and MiSeq output, and were between 0.5–2.27 (Fig. 3B). The average PRN was 1.37, and the $ND_{50}$ and CV of PRNs were consequently 0.99 and 0.25, respectively. No strong correlations were observed between total dRFU values and PRNs after normalisation ($R^2 = 0.25$).

## DISCUSSION

PCR is commonly used as a target enrichment method for massively parallel sequencing (*Hosomichi et al., 2014*; *Hosomichi et al., 2015*). The MspJI-based DNA fragmentation method permits inexpensive sequencing library preparation from amplicon-based templates (*Shinozuka et al., 2015*). A combination of use of these techniques with the

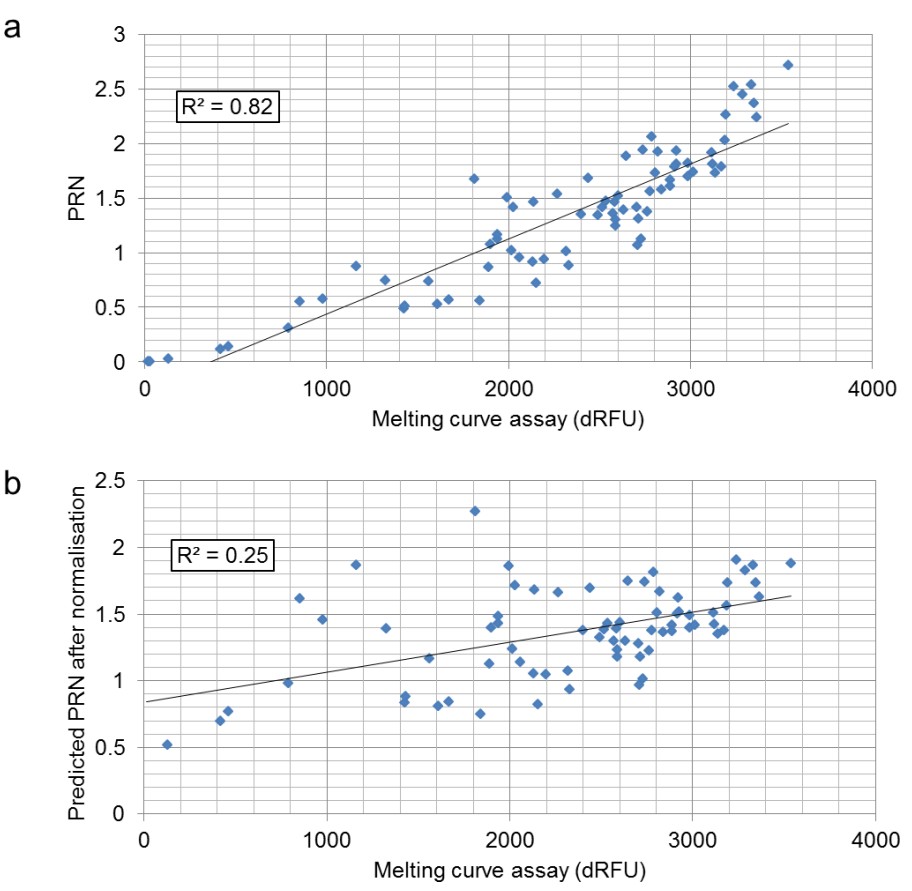

**Figure 3** Correlation between the MC assay-based quantification and PRN of MiSeq sequencing from the 75 samples (A), and between the MC assay-based method and predicted PRN from the library normalisation simulation from the 73 successful samples (B).

MiSeq platform allows low-cost sequencing analysis for the targeted loci. For instance, DNA sequences with a cumulative length of 15 kb could be enriched through PCR with locus-specific primers at a cost of US\$3.4/sample, and a single library could be processed until the library enrichment stage for a cost of US\$3.3 (Table S3). Libraries from up to 384 samples can be sequenced on the MiSeq platform with the current Reagent Kit v3 (600 cycle; Illumina), generating an average of approximately 2,000 times read depth for each nucleotide position within the 15 kb-long regions. As the MiSeq kit costs less than US\$1,500, the per-sample cost for this sequencing option would be under US\$4. Compared with these examples, however, the per-sample costs of the existing high-throughput library normalisation methods, are relatively high (Table 1) even though the primary purpose is solely DNA quantification or normalisation. The MC assay-based library quantification procedure costs substantially less than the other methods; only 1/3 of the NanoDrop-based method. The processing time of the MC assay-based method is also considerably shorter than those of the other methods. For instance, 384 samples may be processed within 3 h on a standard real-time PCR instrument (4 × 96-well microtiter plates), and the duration could be reduced to only 40 min if a 384-well formatted instrument was made available. The

MC assay-based library quantification would, therefore, be most suitable for sequencing experiments typified by large sample number but relatively small-output, such as HLA typing on the MiSeq platform (*Hosomichi et al., 2015*).

As the quantification results from MC assay-, TapeStation-, and NanoDrop-based method were correlated with one other, each method can be used for the purpose of library normalisation. Similar to the previously reported 'over-estimation issue' (*Simbolo et al., 2013*), the quantification results obtained with the NanoDrop system were, however, much larger than those obtained with the other methods, especially when low-concentration libraries were subjected to analysis. Based on the NanoDrop-based method, concentrations of 6.51–9.06 ng/µl were calculated from 4 library samples, from which no DNA was detected with the TapeStation-based method. Similar results were observed when the result of NanoDrop-based quantification was compared with that of the MC assay. As the quantification result from the blank sample (10 mM Tris–HCl buffer without DNA) on the NanoDrop system was $0.17 \pm 0.25$ ng/µl (3 replications), residual molecules from the SPRI-based DNA size-selection may have affected UV-visible spectrophotometry-based quantification.

During the MC assay, the target dsDNA and short DNA can be differentiated on the basis of differences in MT (*Ririe, Rasmussen & Wittwer, 1997*). The preliminary experiment with the short DNA and KAPA fragments suggested that the dRFU values at less than 86 °C broadly represent the quantity of dsDNA shorter than 300 bp in length, while those at and over 86 °C represent the quantity of target DNA. In the subsequent experiments, sequencing libraries with the average fragment size of approximately 500 bp in length were subjected to the MC assay-based quantification, and the correlation between the total dRFU values for the temperature range 86 and 95 °C and other quantification method was demonstrated. As the values between 75 and 95 °C exhibited a lower level of correlation, exclusion of the fluorescence signals from short dsDNA is essential for accurate library quantification (Figs. S4 and S5). The average fragment size of the sequencing library is determined in practice by multiple factors, such as types of the sequencing platforms, sequencing reagents, library preparation methods and experiment purposes. Optimisation of the temperature range of the MC assay, therefore, may be required, depending on the average fragment size of the sequencing library.

The length of DNA fragments is not the only factor determining the MT point, as the GC-content ratio and sequence complexity are also relevant (*Ririe, Rasmussen & Wittwer, 1997*). Differences between sequencing libraries for these parameters may hence influence accuracy of the MC assay-based quantification, and the method may not be suitable for a comparison of DNA quantities, if the DNA sequence contents are considerably different between the libraries. In general, libraries for multiplexed sequencing are, however, prepared in parallel from the identical targeted sequence(s) (*Campbell, Harmon & Narum, 2015*; *Hosomichi et al., 2014*; *Shinozuka et al., 2015*), and the DNA sequence contents of libraries should not be substantially different. For the scenario of different DNA sequence contents, NanoDrop- or TapeStation-based methods would be more suitable.

The CV of PRNs and $ND_{50}$ from the first normalisation experiment were 0.31 and 0.98, respectively, and those from the second simulation experiment were 0.25 and 0.99,

respectively. The result of the MC assay-based method may hence be almost equivalent to that of the BeNUS (Bead-based Normalisation for Uniform Sequencing depth) method (CV of PRNs = 0.31, $ND_{50}$ = 0.93) (*Hosomichi et al., 2014*), although the normalisation result obtained with the qMiSeq method (CV of PRNs = 0.05) was superior to the current result (*Katsuoka et al., 2014*).

In this study, relatively accurate quantification results with the MC assay-based method have been shown. Although the SYBR Green I fluorescent dye has been commonly used for a real-time PCR and MC assay in molecular biology laboratories, the Eva Green (Biotium, CA, USA) and SYTO fluorescence dyes (Thermo Fisher Scientific) may provide better performance than SYBR Green I (*Eischeid, 2011*). Use of such fluorescence dyes may improve the library quantification performance of the MC-based method.

The BeNUS method permits high-throughput automated library normalisation, through reduction of the amount of SPRI beads necessary to capture a limited amount of DNA (*Hosomichi et al., 2014*). This method may consequently be ineffective when library concentrations are too low. The results of the current study suggest that the MC assay-based method performs better for recovery of libraries with low concentration. Due to the high-throughput nature of both the BeNUS and MC assays, a combination of the two may provide a high-quality automated normalisation procedure suitable for an even larger sample volume, numbering in the several thousands.

## ACKNOWLEDGEMENTS

The authors would like to thank Dr Emma Ludlow for provision of the perennial ryegrass-associated fungal endophyte materials, and Dr Noel Cogan for technical advice.

### Funding

This work was supported by funding from the Victorian Department of Economic Development, Jobs, Transport and Resources. The funders had no role in study design, data collection and analysis, decision to publish, or preparation of the manuscript.

### Grant Disclosures

The following grant information was disclosed by the authors:
The Victorian Department of Economic Development, Jobs, Transport and Resources.

### Competing Interests

The authors declare there are no competing interests.

### Author Contributions

- Hiroshi Shinozuka conceived and designed the experiments, performed the experiments, analyzed the data, contributed reagents/materials/analysis tools, wrote the paper, prepared figures and/or tables, reviewed drafts of the paper.
- John W. Forster conceived and designed the experiments, wrote the paper, reviewed drafts of the paper.

## Data Availability

The research in this article did not generate any raw data.

## Supplemental Information

Supplemental information for this article can be found online at http://dx.doi.org/10.7717/peerj.2281#supplemental-information.

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
