# Peer review of "Use of the melting curve assay as a means for high-throughput quantification of Illumina sequencing libraries"

_PeerJ, doi:10.7717/peerj.2281_

## Round 0.1 · original submission · Major Revisions

· Academic Editor

Major Revisions

Your work addresses a timely issue. Alternatives for cost and time-saving, but still high-throughput normalization of NGS libraries represent an urgent need in the sequencing community. I would like to add some considerations to the comments of the reviewers:

- Many researchers are not so familiar with MCA-based quantification, as they are e.g. with qPCR based quantification. In order to make this technique easily accessible to as many researchers as possible, please clarify and expand the technical part by explaning more in detail and/or exemplifying how library quantification and normalization by dRFU is done "in practice". This will support researchers in the field who are not so familiar with MCA.

- To ensure reproducibility, please describe more in detail the origin, length and/or sequence of the Y-shaped adaptors used for MCA calibration.

- It is not strictly necessary, but you might consider moving some parts of the Discussion in a final “Conclusion” section (see the “Instruction for authors”).

Reviewer 1 ·

Basic reporting

No comments

Experimental design

No comments

Validity of the findings

No comments

Additional comments

Line 110 ...(10) is probably not the correct citation style of the Journal!
I would like to see a graphical comparison between the described method and the standard methods - to better show the advantages - reduction of costs and time!

Reviewer 2 ·

Basic reporting

The manuscript is well done: the English language is appropriate and clear. The description of the background, methods, results and discussion are clear and easy to read and understand.

Experimental design

The study evaluates a cheaper and easy method for the evaluation of libraries for Next generation performed by Illumina.

The authors compared 3 different combination of three different PCR conditions associated or not to library normalization by three different procedures.

Validity of the findings

The results are promising for developing easy and cheaper procedure for Normalization of multiplexed libraries.

Additional comments

Few minor improvements should be done:

Results:

Comparison of the library quantification methods
Please, add at the beginning of this paragraph an introductive sentence describing the approaches used for comparing the quantification methods consequently put here the citation of Supplementary figure S2.

The position of the reference to Supplementary Figure S2 at the end of the second sentence in this paragraph is not appropriate: in this sentence the authors did not explain the figure and conversely in the figure they did not explain only the PCR enrichment, but the different approaches (PCR conditions/normalization procedure) used in the setting of the study.

Question:

The authors could kindly explain the use of GYBR Green 1 and they did not used dyes of third generation (i.e. Eva Green, Syto 9)? Does the SYBRGreen give the sufficient accuracy for the evaluation of melting temperature in the analysed amplicons?

Reviewer 3 ·

Basic reporting

This manuscript addresses an important technical issue in NGS library preparation for multi-samples. The authors are carefully treating the advantage of the normalisation method focused on both performance and cost. The platform is based on well-established real-time PCR-based method and commonly used.

Experimental design

The focus of the paper is easy to understand and the method worked well for PRN, but the data of library quality is not clear. dRFU was used to control PRN, but heterogeneity among libraries was not considered well. For example, TapeStation chromatogram is useful to understand status of the library.

Validity of the findings

No Comments

Additional comments

The quality of reads such as duplication rate, coverage of target region, read depth of high GC% region should be discussed in the manuscript, because importance of quantification method for sequencing libraries is not only PRN and cost.

---

## Round 0.2 · accepted · Accept

· Academic Editor

Accept

Dear Hiroshi

I am glad to inform you, that the revised version of the manuscript is now acceptable for publication. I would also like to take the opportunity to apologize for the long review time, which has been caused by an unexpectedly long search for reviewers. Thank you for your understanding, and, once again, congratulations.

Reviewer 3 ·

Basic reporting

No Comments

Experimental design

No Comments

Validity of the findings

No Comments

Additional comments

The revised version of the paper is much improved with focusing the advantage of MCA-based quantification method carefully. As follow-up story for previous paper (Shinozuka et al., 2015), authors refered fine tuning of the method, especially for cost and processing time. This paper is an important contribution and acceptable for publication.